# Phylogeny of the Chinese Subgenera of the Genus *Homoneura* (Diptera, Lauxaniidae, Homoneurinae) Based on Morphological Characters

**DOI:** 10.3390/insects13080665

**Published:** 2022-07-22

**Authors:** Chaoyang Kong, Keli Feng, Shengjuan Zhao, Wenliang Li, Xuankun Li

**Affiliations:** 1College of Horticulture and Plant Protection, Henan University of Science and Technology, Luoyang 471023, China; 200320261315@stu.haust.edu.cn (C.K.); 201419030202@stu.haust.edu.cn (K.F.); 2College of Food & Bioengineering, Henan University of Science and Technology, Luoyang 471023, China; zshj-114@163.com; 3Department of Entomology, College of Plant Protection, China Agricultural University, Beijing 100193, China; 4Department of Biological Sciences, University of Memphis, Memphis, TN 38152, USA; 5Center for Biodiversity Research, University of Memphis, Memphis, TN 38152, USA

**Keywords:** phylogeny, morphological characters, *Homoneura*, subgenus, China

## Abstract

**Simple Summary:**

The genus *Homoneura* is the most speciose genus of Lauxaniidae. However, no phylogenetic study of *Homoneura* has been published using morphological data, despite the high diversity and ecological significance. Therefore, we present the first morphological phylogeny of *Homoneura*. The monophyly of the genus *Homoneura* and the subgenus *Homoneura* is not supported. The monophyly of the subgenera *Euhomoneura* and *Neohomoneura* is supported, as well as the sister relationship between the subgenera *Chaetohomoneura* and *Neohomoneura*. These results provide a comprehensive framework and information toward future research of this genus.

**Abstract:**

The genus *Homoneura* comprises over 700 described species in eight known subgenera distributed worldwide and has the highest species richness of Lauxaniidae. Five subgenera and more than 200 species have currently been recorded from China. Despite its high diversity, the monophyly of *Homoneura* and its subgenera, and the phylogenetic relationships among its subgenera remain to be investigated. One maximum-parsimony tree was generated based on 105 morphological characters scored from 24 species, representing all five subgenera of *Homoneura* recorded from China. The results did not support the monophyly of the genus *Homoneura* and subgenus *Homoneura*. The subgenus *Chaetohomoneura* is a sister to subgenus *Neohomoneura.* The monophyly of the subgenera *Euhomoneura* and *Neohomoneura* is supported. Much of the current classification of the genus *Homoneura* needs a revision before taxonomy can reflect natural groupings.

## 1. Introduction

The family Lauxaniidae is species-rich and globally distributed, with three subfamilies, more than 170 genera, and nearly 2100 described species [1]. Lauxaniids have a variety of habitats [2,3], mainly scavenging and fungus feeding [4,5], and have the habit of visiting flowers [6,7]. They play a very important role in the ecological system, degrading organic matter, plant pollination, and maintaining ecological balance [8,9,10]. In addition, lauxaniids are sensitive to environmental change. They have been used as an indicator organism for environmental change assessments of farmland ecosystems in Europe and have also become one of the research hotspots of terrestrial ecosystem assessment indicators [11].

The genus *Homoneura* is the most speciose genus of Lauxaniidae. Currently, it contains eight subgenera and more than 700 described species worldwide, among which five subgenera and more than 200 species are recorded from China [1]. It is also one of the most diverse genera of the Acalyptratae. *Homoneura* is widely distributed in all major animal geographical areas except for the Neotropical region.

Stuckenberg hypothesized the relationship among 19 genera of the subfamily Homoneurinae based on morphological characters and divided 19 genera into three groups [12]. Kim used numerical methods to cluster Australian members of the genera *Homoneura*, *Trypeisoma*, and allied genera by their similarities in morphology [13]. In the only existing molecular phylogenetic study of the generic level relationships of Lauxaniidae, Shi et al. explored the phylogeny of the subgenus *Homoneura* based on two mitochondrial and two nuclear genes [14]. The monophyly of the subgenus *Homoneura* was not recovered. No phylogenetic study of the genus *Homoneura* has been published using morphological data. Despite the high diversity and ecological significance, the monophyly of the genus *Homoneura* and its subgenera, and the phylogenetic relationships among the subgenera remain to be investigated.

In this study, we use 105 morphological characters to reconstruct the phylogeny based on all five subgenera of *Homoneura* that are recorded from China, aiming to (1) test the monophyly of the genus *Homoneura*; (2) test the monophyly of Chinese subgenera of *Homoneura*; (3) investigate relationships among five Chinese subgenera of *Homoneura*.

## 2. Materials and Methods

### 2.1. Morphological Study and Terminology

General terminology follows Cumming & Wood and Gaimari & Silva [15,16]. Genitalia preparations were made by removing and macerating the apical portion of the abdomen in cold saturated NaOH for six hours, then rinsing and neutralizing them for dissection and study. After examination in glycerine, they were transferred to fresh glycerine and stored in a microvial pinned below the specimen or moved to an ethanol tube together with the wet specimens. Most characters were illustrated using photographs and line drawings. Photographs were taken using a Canon EOS6D microscope (Canon, Tokyo, Japan) and stacked using HELICO FOCUS v7.0.2.0 (Helicon Soft, Kharkiv, Ukraine). Line drawings were drawn with Adobe Illustrator 2021 v25.2.1 (Adobe, San Jose, CA, USA).

### 2.2. Specimens Examined and Morphological Characters

In total, 24 lauxaniid species were selected in the analysis, including 17 *Homoneura* species representing five subgenera. Two species of Lauxaniinae: *Minettia* (*frendelia*) *longipennis* and *Pachycerina decemlineata* (Figure 1C), and five species of Homoneurinae: *Cestrotus liui* (Figure 1B), *Dioides incurvatus*, *Noonamyia umbrellata* (Figure 1A), *Phobeticomyia motuoensis*, and *Prosopophorella yoshiyasui* were selected as outgroup taxa. Appendix A shows the terminals included in the cladistic analysis.

On the basis of our survey, 105 morphological characters obtained from adults from the head (28 characters, Figure 2), thorax (10 characters, Figure 3A–F), legs (9 characters, Figure 3G–L), wing (17 characters, Figure 4), abdomen (7 characters, Figure 5), male genitalia (29 characters, Figure 6, Figure 7 and Figure 8), and female genitalia (5 characters, Figure 9) were numerically coded (Appendix A). Eighty-seven characters are binary and 18 are multistate. All characters were treated as unordered and with equal weight. Missing character states were coded with (?), and inapplicable states were scored as (–).

The studied specimens are deposited in the Insect Collection of Henan University of Science and Technology (HAUST).

**Figure 1 insects-13-00665-f001:**
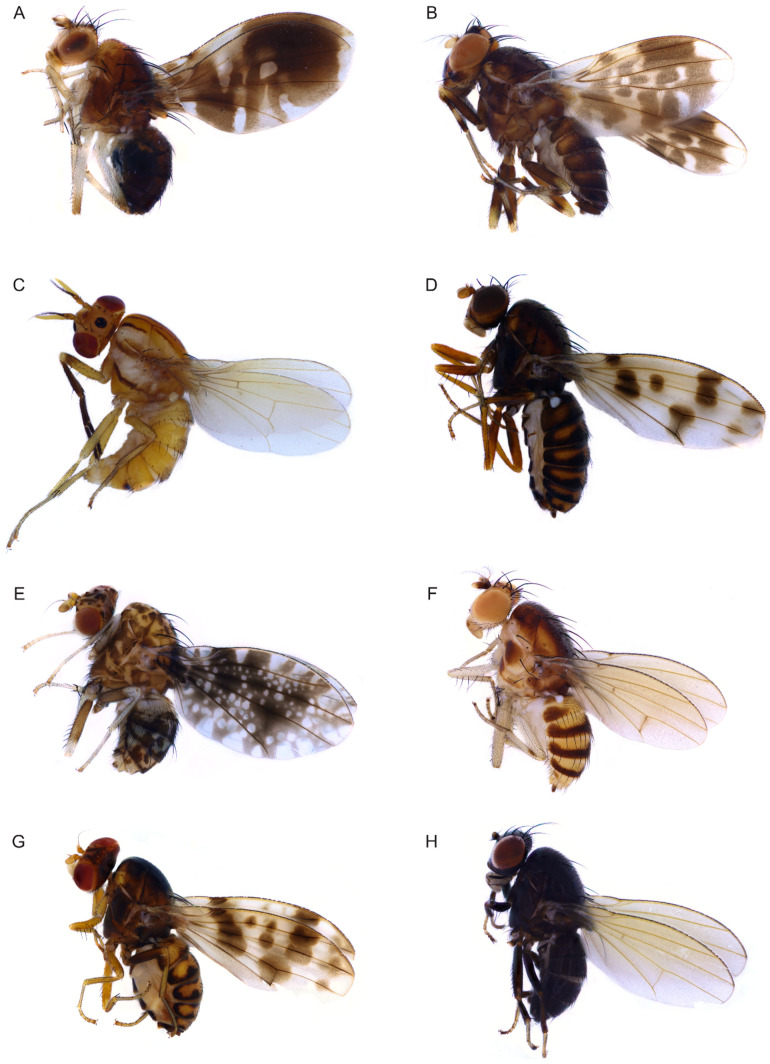
Adults of species of Lauxaniidae. (**A**): *Noonamyia umbrellata*; (**B**): *Cestrotus liui*; (**C**): *Pachycerina decemlineata*; (**D**): *Homoneura* (*Euhomoneura*) *shatalkini*; (**E**): *Homoneura* (*Homoneura*) *picta*; (**F**): *Homoneura* (*Homoneura*) *flavida*; (**G**): *Homoneura* (*Homoneura*) *dorsacerba*; (**H**): *Homoneura* (*Homoneura*) *noticomata*.

### 2.3. List of Characters Used in the Cladistic Analysis

Head:Shape of head, height of head/width of head: (0) > 4/5 (Figure 2J); (1) ≤ 4/5 (Figure 2E).Color of head: (0) black (Figure 2L); (1) brown to yellow (Figure 2D).Color of ocellar triangle: (0) black (Figure 2L); (1) brown to yellow (Figure 2G).Length of ocellar seta/length of anterior fronto-orbital seta: (0) ≥ 1 (Figure 2N); (1) < 1 (Figure 2E).Length of anterior fronto-orbital seta/length of posterior fronto-orbital seta: (0) < 1 (Figure 2B); (1) ≥ 1 (Figure 2M).Frons: (0) flat; (1) uplifted; (2) sunken.Length of frons/width of frons: (0) ≥ 1 (Figure 2E); (1) < 1 (Figure 2A).Middle of frons: (0) without spot or stripe (Figure 2G); (1) with a dark median longitudinal stripe extending from anterior margin to ocellar triangle (Figure 2C).Between the middle of frons and fronto-orbital seta: (0) without spot or stripe (Figure 2D); (1) with two longitudinal stripes extending to both sides of ocellar triangle (Figure 2F); (2) with a spot (Figure 2H).Base of fronto-orbital seta: (0) without spot or srtipe (Figure 2H); (1) with spot (Figure 2A); (2) with stripe (Figure 2B).Spot on anterior margin of frons: (0) absent; (1) present.Color of face: (0) black (Figure 2P); (1) brown to yellow (Figure 2I).Spot on face: (0) absent (Figure 2J); (1) present (Figure 2M).Middle of face: (0) flat (Figure 2N); (1) uplifted (Figure 2M).Ventral margin of face: (0) flat (Figure 2O); (1) uplifted (Figure 2K).Width of ventral margin of face/height of gena: (0) ≥ 3X; (1) < 3X.Spot on gena: (0) absent (Figure 2O); (1) present (Figure 2K).Below eye on gena: (0) without strong seta (Figure 2P); (1) with strong seta (Figure 2H).Length of gena/length of eye: (0) ≤ 1; (1) > 1.Color of pedicel: (0) yellow or brown (Figure 2P); (1) black (Figure 2H).Pedicel: (0) without two strong ventral setae (Figure 2I); (1) with two strong ventral setae (Figure 2J).Color of flagellum: (0) monochrome (Figure 2O); (1) bicolor (Figure 2N).Arista: (0) plumose (pubescent with longest setulae, not less than 1/3 the width of the first flagellomere) (Figure 2M); (1) pubescent (pubescent with longest setulae shorter than 1/3 the width of the first flagellomere) (Figure 2F).Length of the first flagellomere/width of the first flagellomere: (0) ≥ 2X (Figure 2P); (1) < 2X (Figure 2N).Between base of antennae and inner margin of eye: (0) with spot (Figure 2H); (1) without spot (Figure 2D).Color of proboscis: (0) black; (1) brown to yellow (Figure 2N).Color of palpus: (0) monochrome, black (Figure 2L); (1) monochrome, brown (Figure 2K); (2) bicolor.Wide band on occiput; (0) absent (Figure 2G); (1) present (Figure 2C).

**Figure 2 insects-13-00665-f002:**
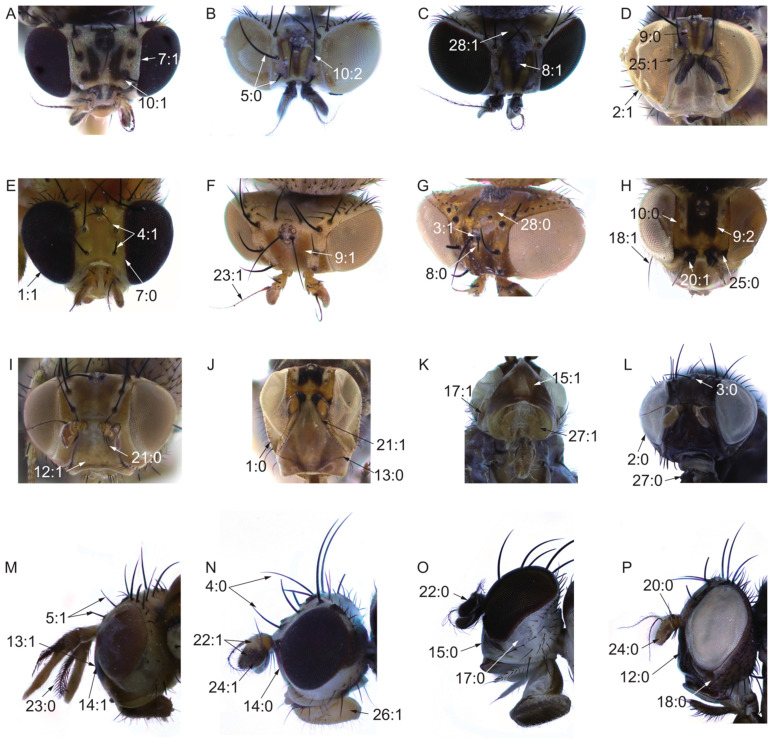
Hand characters. (**A**,**N**): *Homoneura* (*Homoneura*) *picta*; (**B**,**C**,**D**,**O**): *Homoneura* (*Homoneura*) *flavida*; (**E**): *Homoneura* (*Minettioides*) *orientis*; (**F**,**I**): *Homoneura* (*Euhomoneura*) *yanqingensis*; (**G**): *Homoneura* (*Neohomoneura*) *tricuspidata*; (**H**,**J**,**K**): *Prosopophorella yoshiyasui*; (**L**,**P**): *Minettia* (*Frendelia*) *longipennis*; (**M**): *Pachycerina decemlineata*.

Thorax:29.Color of mesonotum: (0) black (Figure 3A); (1) brown to yellow (Figure 3E).30.Stripe on mesonotum: (0) present (Figure 3C); (1) absent (Figure 3E).31.Base of dorsocentral seta and prescutellar acrostichal seta: (0) without spot (Figure 3D); (1) with spot (Figure 3B).32.Pre-sutural dorsocentral seta; (0) absent (Figure 3A); (1) present (Figure 3D).33.Post-sutural dorsocentral seta: (0) three, 1st post-sutural dorsocentral setae behind the transversal suture (Figure 3E); (1) three, 1st post-sutural dorsocentral setae in the transversal suture; (2) two (Figure 3D).

**Figure 3 insects-13-00665-f003:**
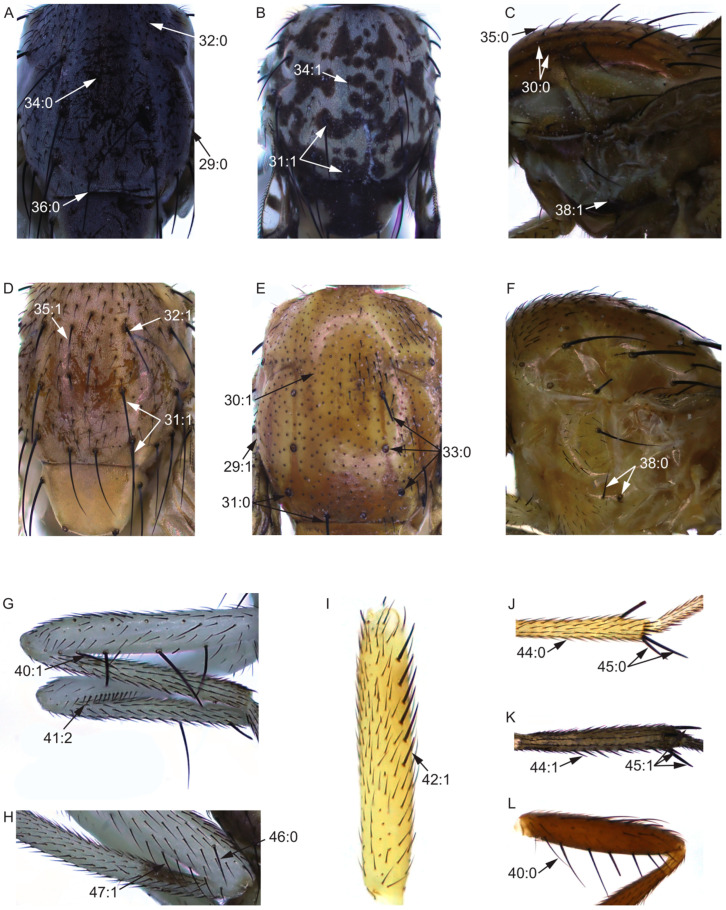
Thoracic and leg characters. (**A**,**G**,**H**): *Homoneura* (*Homoneura*) *flavida*; (**B**): *Homoneura* (*Homoneura*) *picta*; (**C**): *Pachycerina decemlineata*; (**D**,**I**,**J**): *Homoneura* (*Euhomoneura*) *yanqingensis*; (**E**,**K**): *Homoneura* (*Neohomoneura*) *tricuspidata*; (**F**): *Homoneura* (*Minettioides*) *orientis*; (**L**): *Minettia* (*Frendelia*) *longipennis*.

34.Rows of acrostichal seta: (0) no less than seven rows (Figure 3A); (1) less than seven rows (Figure 3B).35.Acrostichal seta: (0) weak, short hair (Figure 3C); (1) strong seta (Figure 3D).36.Prescutellar acrostichal seta: (0) present (Figure 3A); (1) absent.37.Intra-alar seta: (0) present; (1) absent.38.Katepisternal seta: (0) two (Figure 3F); (1) one (Figure 3C).

Leg:39.Length of leg/length of body: (0) ≤ 1; (1) > 1.40.Posterior ventral seta in fore femur: (0) no less than five (Figure 3L); (1) less than five (Figure 3G).41.Ctenidium short seta in fore femur: (0) absent; (1) no more than ten; (2) more than ten (Figure 3G).42.Anterior seta in mid femur: (0) more than six; (1) no more than six (Figure 3I).43.Posterior ventral seta in mid femur: (0) absent; (1) present.44.Posterior seta in mid tibia: (0) absent (Figure 3J); (1) present (Figure 3K).45.Strong apical ventral seta in mid tibia: (0) no more than two (Figure 3J); (1) three (Figure 3K); (2) four.46.Preapical anterior dorsal seta in hind femur; (0) present (Figure 3H); (1) absent.47.Anteroventral seta in hind femur: (0) absent; (1) present (Figure 3H).

Wing:48.Length of wing/width of wing: (0) < 2.7X (Figure 4B); (1) ≥ 2.7X (Figure 4D).49.Area of transparent area or light-yellow area of wing/area of wing spot area: (0) > 1; (1) ≤ 1.50.Area of transparent or light-yellow area above wing R_4+5_/area of wing spot above R_4+5_: (0) > 1; (1) ≤ 1.51.Spot on R_2+3_: (0) absent (Figure 4G); (1) no longer than half the length of R_2+3_ (Figure 4F); (2) longer than half the length of R_2+3_ (Figure 4B).52.Tip of R_4+5_: (0) without spot (Figure 4E); (1) with one spot and it is not longer than half of the top of R_4+5_ (Figure 4H); (2) with two or more spots and it is not longer than half of the top of R_4+5_; (3) longer than half of the top of R_4+5_ (Figure 4C).53.Spot on crossvein r-m: (0) absent (Figure 4E); (1) present (Figure 4H).54.Spot on crossvein dm-cu: (0) absent (Figure 4D); (1) present (Figure 4F).55.Tip of M_1_: (0) without spot (Figure 4G); (1) with one spot and it is no longer than half of the top of M_1_ (Figure 4C); (2) with one spot and it is longer than half of the top of M_1_ (Figure 4B).56.Stripe on penultimate section of CuA_1_: (0) absent (Figure 4E); (1) present (Figure 4D).57.Spot on base of radial vein and medial vein: (0) absent (Figure 4E); (1) present (Figure 4A).58.Anal vein: (0) present (Figure 4F); (1) absent (Figure 4B).59.Spot on anterior cubital cell: (0) absent (Figure 4G); (1) present (Figure 4A).60.2nd (between R_1_ and R_2+3_) section/3rd (between R_2+3_ and R_4+5_) section: (0) ≥ 3X; (1) < 3X.61.3rd (between R_2+3_ and R_4+5_) section/4th (between R_4+5_ and M_1_) section: (0) ≥ 1.5X; (1) < 1.5X.62.Length of the ultimate section of CuA_1_/length of the penultimate section of CuA_1_; (0) < 1/5; (1) ≥ 1/5.63.Crossvein r-m: (0) before or in the middle of the discal cell (Figure 4F); (1) behind the middle of the discal cell (Figure 4D).64.Color of knob part of haltere: (0) black; (1) brown or yellow.

**Figure 4 insects-13-00665-f004:**
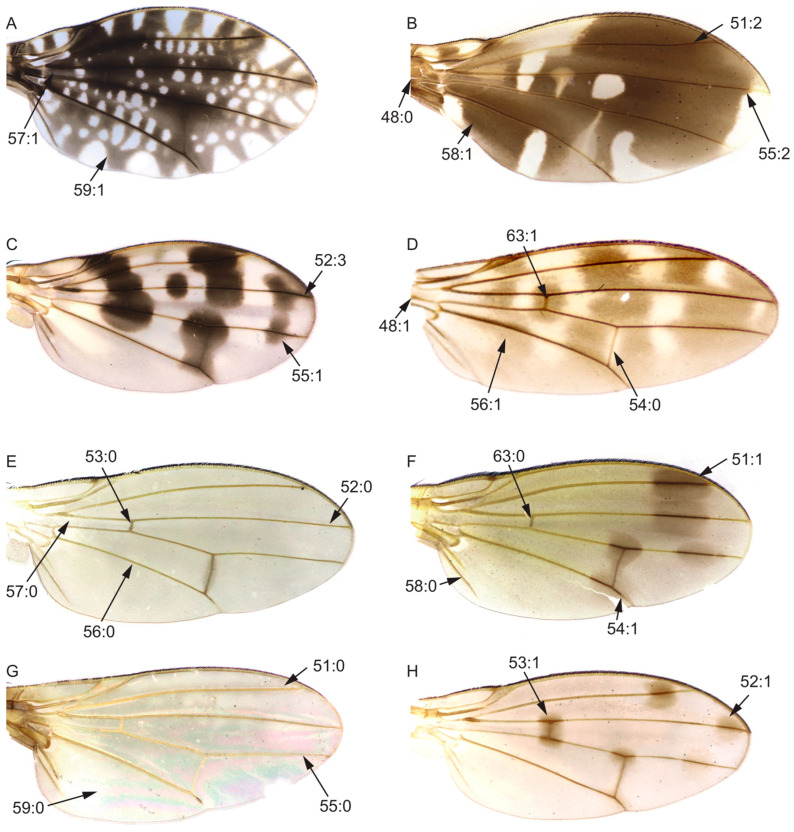
Wing characters. (**A**): *Homoneura* (*Homoneura*) *picta*; (**B**): *Noonamyia umbrellata*; (**C**): *Homoneura* (*Homoneura*) *posterotricuspis*; (**D**): *Prosopophorella yoshiyasui*; (**E**): *Homoneura* (*Homoneura*) *flavida*; (**F**): *Homoneura* (*Neohomoneura*) *zengae*; (**G**): *Minettia* (*Frendelia*) *longipennis*; (**H**): *Homoneura* (*Euhomoneura*) *yanqingensis*.

Abdomen:65.Color of tergites 3–6: (0) black; (1) brown or yellow.66.Spot on middle of tergite 2: (0) absent (Figure 5B); (1) present (Figure 5A).67.Spot on side of tergite 2: (0) absent (Figure 5E); (1) present (Figure 5D).68.Spot on middle of tergite 5: (0) absent (Figure 5C); (1) present (Figure 5D).69.Side of tergite 5: (0) without spot (Figure 5F); (1) with striped spot (Figure 5D); (2) with circular spot (Figure 5E).70.Spot on middle of tergite 6: (0) absent (Figure 5C); (1) present (Figure 5A).71.Spot on side of tergite 6: (0) absent (Figure 5F); (1) present (Figure 5D).

**Figure 5 insects-13-00665-f005:**
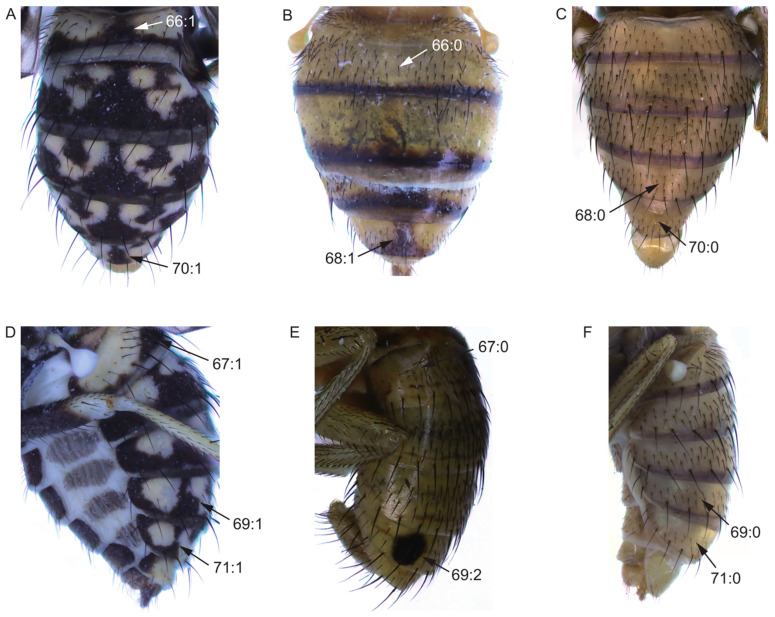
Abdomen characters. (**A**,**D**): *Homoneura* (*Homoneura*) *picta*; (**B**): *Homoneura* (*Neohomoneura*) *tricuspidata*; (**C**,**F**): *Homoneura* (*Euhomoneura*) *yanqingensis*; (**E**): *Homoneura* (*Minettioides*) *orientis*.

Male genitalia:72.Shape of syntergosternite: (0) semicircular (Figure 6B); (1) circular, without ventral processes (Figure 6A); (2) circular, with ventral processes (Figure 6C).73.Length of dorsal margin of syntergosternite/length of posterior margin of syntergosternite: (0) < 1/2; (1) ≥ 1/2.74.Dorsal margin of syntergosternite: (0) without short hair (Figure 6B); (1) with short hair (Figure 6C).75.Syntergosternite around the spiracle: (0) without short hair (Figure 6B); (1) with short hair (Figure 6A).76.Length of dorsal margin of epandrium/length of ventral margin of epandrium: (0) > 1/2; (1) ≤ 1/2.

**Figure 6 insects-13-00665-f006:**
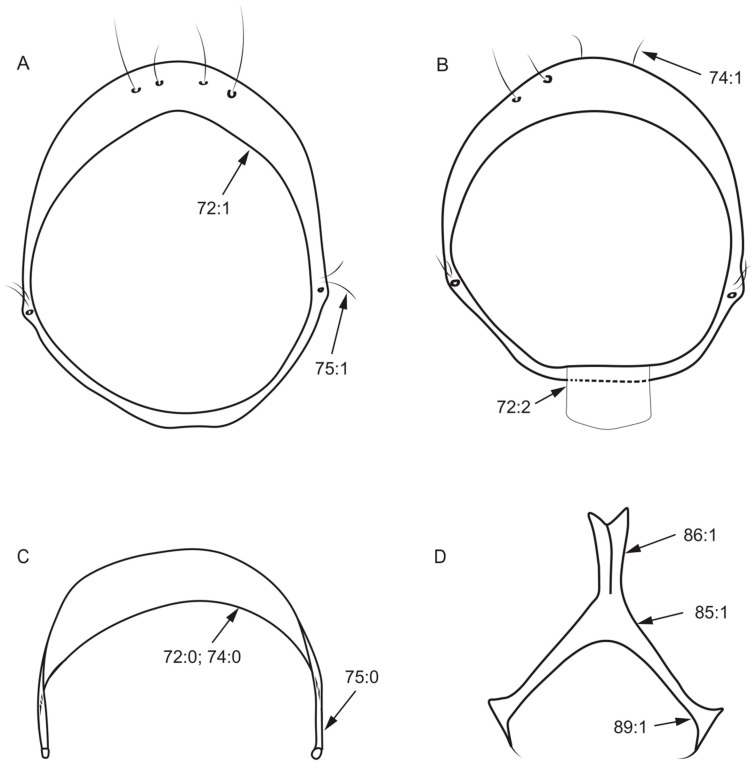
Syntergosternite and hypandrium characters. (**A**): *Homoneura* (*Homoneura*) *dorsacerba*; (**B**): *Homoneura* (*Homoneura*) *posterotricuspis*; (**C**): *Homoneura* (*Homoneura*) *procerula*; (**D**): *Cestrotus liui*.

77.Surstylus: (0) separated from epandrium (Figure 7D); (1) not separated from epandrium (Figure 7B).78.Number of surstylus processes: (0) one (Figure 7D); (1) two or more (Figure 7C).79.Length of the longest surstylus/height of epandrium: (0) ≥ 1/2 (Figure 7A); (1) < 1/2 (Figure 7B).80.Shape of the apex of surstylus: (0) sharp (Figure 7A); (1) blunt (Figure 7C).81.Shape of surstylus: (0) bent (Figure 7A); (1) straight (Figure 7B).

**Figure 7 insects-13-00665-f007:**
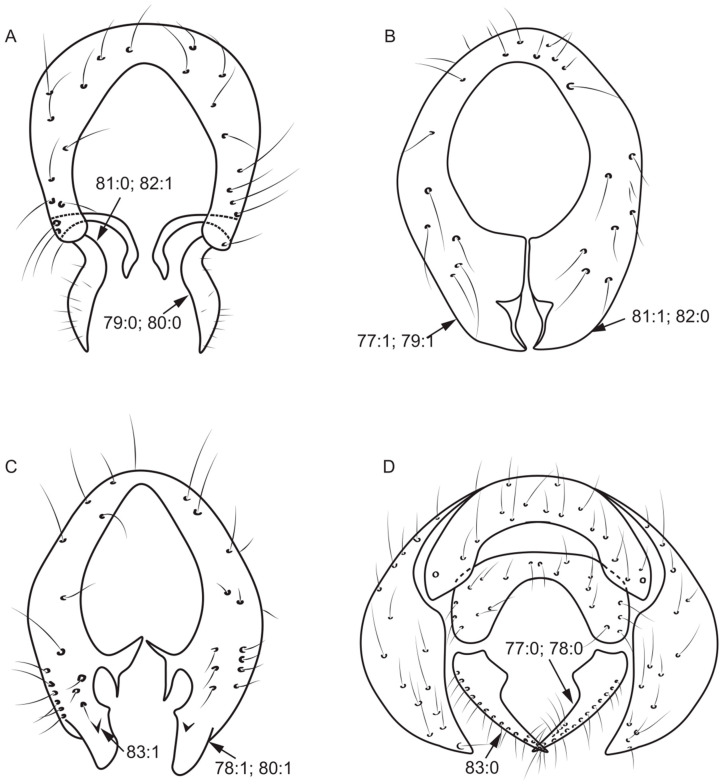
Epandrium characters. (**A**): *Homoneura* (*Homoneura*) *trispina*; (**B**): *Homoneura* (*Homoneura*) *dorsacerba*; (**C**): *Homoneura* (*Homoneura*) *posterotricuspis*; (**D**): *Pachycerina decemlineata*.

82.Width of the middle of surstylus/length of surstylus: (0) ≥ 1/2 (Figure 7B); (1) < 1/2 (Figure 7A).83.Fine teeth or terminal processes on surstylus: (0) absent (Figure 7D); (1) present (Figure 7C).84.Surstylus: (0) without seta; (1) with seta.85.Shape of hypandrium: (0) U-shaped (Figure 8A); (1) Y-shaped (Figure 6D); (2) H-shaped (Figure 8D); (3) W-shaped (Figure 8B).86.Middle of anterior margin of hypandrium: (0) without inner processes (Figure 8D); (1) with inner processes (Figure 6D).87.Both sides anterior margin of hypandrium: (0) without inner processes; (1) with inner processes (Figure 8A).88.Middle of posterior margin of hypandrium: (0) without ventral process (Figure 8A); (1) with ventral process (Figure 8D).

**Figure 8 insects-13-00665-f008:**
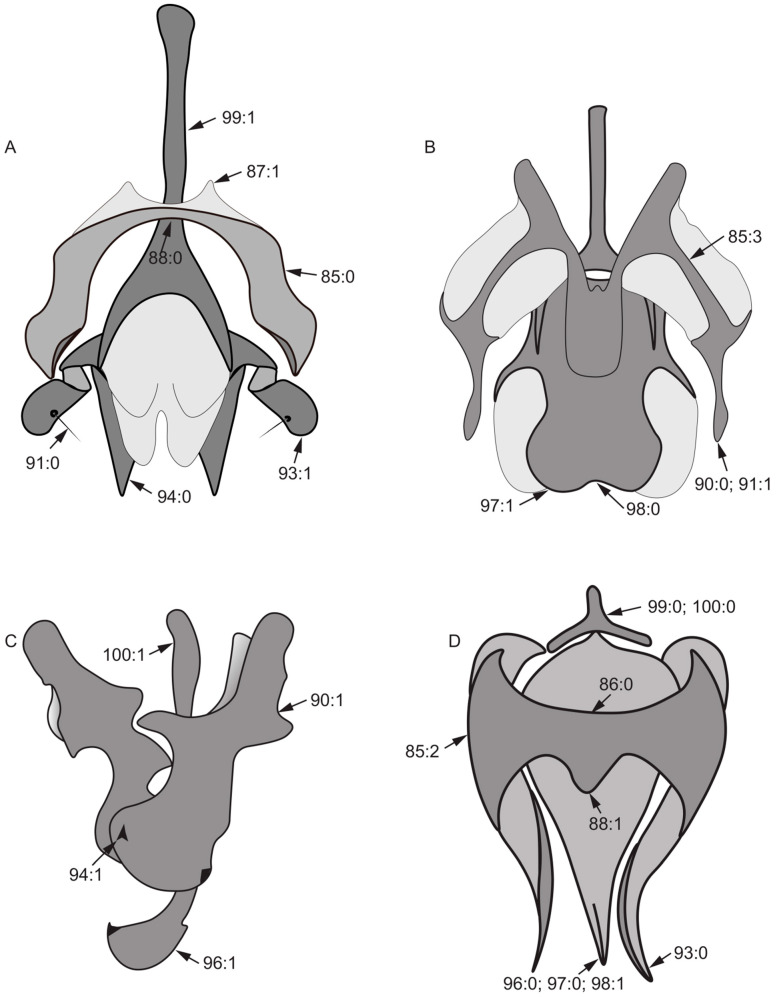
Aedeagal complex characters. (**A**): *Noonamyia umbrellata*; (**B**): *Homoneura* (*Homoneura*) *beckeri*; (**C**): *Pachycerina decemlineata*; (**D**): *Homoneura* (*Homoneura*) *procerula*.

89.Both sides posterior margin of hypandrium: (0) without ventral process; (1) with ventral process (Figure 6D).90.Gonite: (0) present (Figure 8B); (1) absent (Figure 8C).91.Seta on gonite: (0) present (Figure 8A); (1) absent (Figure 8B).92.Length of gonite/length of phallus: (0) ≥ 1/2; (1) < 1/2.93.Tip of gonite: (0) sharp (Figure 8D); (1) blunt (Figure 8A).94.Phallus: (0) without thorn or sharp process (Figure 8A); (1) with thorn or sharp process (Figure 8C).95.Lateral view of the top of phallus: (0) bent; (1) straight.96.Tip of phallus: (0) not inflated (Figure 8D); (1) blunt round apically (Figure 8C).97.Apex of phallus: (0) sharp (Figure 8D); (1) not sharp (Figure 8B).98.Phallus: (0) with distinct apical concave (Figure 8B); (1) without distinct apical concave (Figure 8D).99.Length of aedeagal apodeme/length of phallus: (0) < 1 (Figure 8D); (1) ≥ 1 (Figure 8A).100.Width of the base of aedeagal apodeme/width of the middle of aedeagal apodeme: (0) > 2X (Figure 8D); (1) ≤ 2X (Figure 8C).

Female genitalia:101.Number of spermathecae: (0) 1 + 2 (Figure 9A); (1) 2 + 2 (Figure 9B).102.Shape of spermathecae: (0) globular (Figure 9B); (1) droplet-shaped (Figure 9C); (2) tubular (Figure 9D).103.Spermathecae: (0) connected with spermathecal duct via a rod (Figure 9A); (1) connected with spermathecal duct directly (Figure 9C).104.Both sides of posterior margin of sternite 8: (0) with process; (1) without process.105.Posterior margin of tergite 9: (0) uplifted; (1) sunken.

**Figure 9 insects-13-00665-f009:**
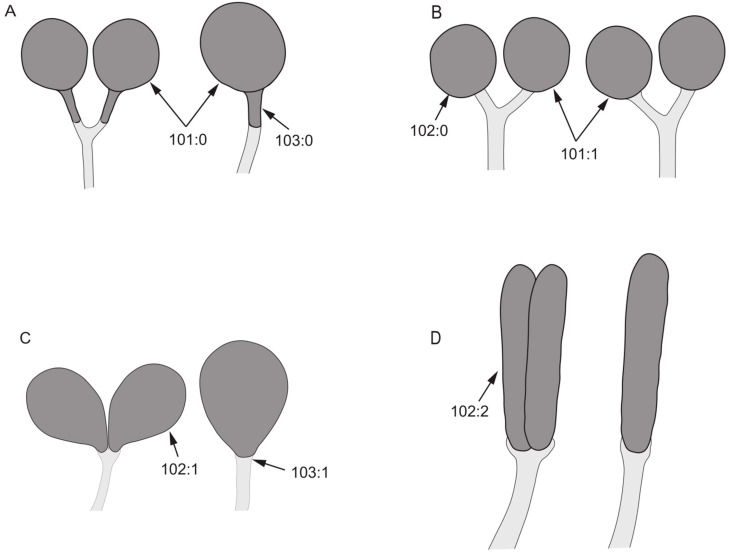
Spermathecae characters. (**A**): *Minettia* (*Frendelia*) *longipennis*; (**B**): *Noonamyia umbrellata*; (**C**): *Cestrotus liui*; (**D**): *Homoneura* (*Minettioides*) *orientis*.

### 2.4. Cladistic Analysis

The phylogenetic reconstruction was conducted using maximum-parsimony analysis using NONA v2.0 (Goloboff, P.A., Tucuman, Argentina) with a heuristic search by 1000 replications, and TNT with 1000 random-addition traditional searches [17,18]. All characters were initially equally weighted. Branch support values were verified through bootstrap analyses on NONA 2.0 with 100 replications. The Bremer support value or decay index for the resultant strict consensus tree was calculated using TNT [19]. The unambiguous characters were mapped on the tree using WinClada version v1.00.08 (Nixon, K.C., Ithaca, NY, USA) [20].

## 3. Results

### 3.1. Phylogenetic Analysis

Maximum-parsimony analyses yielded one maximum-parsimonious tree [branch length = 337, consistency index (CI) = 0.36, retention index (RI) = 0.56]. The maximum-parsimony tree is shown in Figure 10, Bootstrap values (BS) and Bremer support (B) values are presented next to the nodes.

### 3.2. Subfamily Homoneurinae

The monophyly of the subfamily Homoneurinae was supported (BS = −/B = 3) by seven synapomorphies: between the middle of frons and fronto-orbital setae with two spots (9:1), length of the first flagellomere/width of the first flagellomere < 2X (24:1), posterior ventral seta in fore femur no less than five (40:1), ctenidium short seta in fore femur with more than ten (41:2), surstylus not separated from epandrium (77:1), phallus without distinct apical concave (98:1), and width of the base of aedeagal apodeme/width of the middle of aedeagal apodeme ≤ 2X (100:1).

Two major lineages were recovered within Homoneurinae. The clade 1 was supported by one synapomorphy: mesonotum without stripe (30:1), and four homoplasious characters: wing 3rd (between R_2+3_ and R_4+5_) section/4th (between R_4+5_ and M_1_) section < 1.5X (61:1), hypandrium H-shaped (85:2), both sides anterior margin of hypandrium with inner processes (87:1), and lateral view of the top of phallus straight (95:1). The clade 2 was supported by two synapomorphies: crossvein r-m with spot (53:1), and wing 2nd (between R_1_ and R_2+3_) section/3rd (between R_2+3_ and R_4+5_) section < 3X (60:1).

### 3.3. The Genus Homoneura and Allies

The genus *Homoneura* was not recovered as monophyletic, and three graded clades (clades 1, 3, and 4) were detected instead. Clade 1 included subgenera *Minettioides*, *Neohomoneura*, and *Chaetohomoneura*, and part of the subgenus *Homoneura* (*Homoneura* (*H.*) *acrostichali*, *Homoneura* (*H.*) *crassicauda*, *Homoneura* (*H.*) *noticomata*, *Homoneura* (*H.*) *trispina*, *Homoneura* (*H.*) *beckeri*, and *Homoneura* (*H.*) *flavida*). Clade 3 represented subgenus *Euhomoneura* and was supported by two synapomorphies: pre-sutural dorsocentral seta present (32:1), and two post-sutural dorsocentral setae (33:2). Clade 4 included another part of the subgenus *Homoneura* (*Homoneura* (*H.*) *picta*, *Homoneura* (*H.*) *procerula*, *Homoneura* (*H.*) *dorsacerba*, and *Homoneura* (*H.*) *posterotricuspis*), which was supported by one synapomorphy: side of tergite 6 without spot (71:1). 

The other five included the Homoneurinae genera, *Cestrotus*, *Dioides*, *Noonamyia*, *Phobeticomyia*, and *Prosopophorella*, forming a monophyletic group (clade 7) and sister to clade 4 (nested within the genus *Homoneura*). The monophyly of clade 7 was supported by one synapomorphy: length of gonite/length of phallus < 1 (92:1), and two homoplasious characters: length of gena/length of eye > 1 (19:1), and crossvein r-m located behind the middle of the discal cell (63:1).

### 3.4. Monophyly and Relationships among Chinese Subgenera of Homoneura

The monophyly of the subgenera *Neohomoneura* and *Euhomoneura* was supported, while the subgenus *Homoneura* was recovered as polyphyletic (clades 4, 5, and 6). Due to the limited taxon sampling, the monophyly of the subgenera *Minettioides* and *Chaetohomoneura* could not be tested. 

Within clade 1, *Minettioides* + clade 5 was sister to clade 6 + (*Chaetohomoneura* + *Neohomoneura*). The monophyly of *Minettioides* + clade 5 was supported by one synapomorphy: side of tergite 5 with circular spot (69:2), and two homoplasious characters: ocellar triangle brown to yellow (3:1), and length of ocellar seta/length of anterior fronto-orbital seta < 1 (4:1). The sister relationship of *Chaetohomoneura* and *Neohomoneura* was supported by one synapomorphy: mid tibia with posterior seta (44:1), and two homoplasious characters: posterior ventral seta in fore femur no less than five (40:0), and phallus with thorn or sharp process (94:1). The monophyly of *Neohomoneura* was supported by seven homoplasies: ocellar triangle brown to yellow (3:1), spot on R_2+3_ no longer than half the length of R_2+3_ (51:1), tip of R_4+5_ with one spot and it is not longer than half of the top of R_4+5_ (52:1), crossvein dm-cu with spot (54:1), tip of M_1_ with one spot and it is longer than half of the top of M_1_ (55:2), dorsal margin of syntergosternite with short hair (74:1), and syntergosternite around the spiracle with short hair (75:1).

Within clade 2, *Euhomoneura* is sister to clade 4 + clade 7. The monophyly of *Euhomoneura* was supported by pre-sutural dorsocentral seta present (32:1), and two post-sutural dorsocentral setae (33:2). The sister relationship of clade 4 and clade 7 was supported by two synapomorphies: area of transparent area or light-yellow area of wing/area of wing spot area ≤ 1 (49:1), and area of transparent or light-yellow area above wing R_4+5_/area of wing spot above R_4+5_ ≤ 1 (50:1). The monophyly of clade 4 was supported by side of tergite 6 without spot (71:1).

## 4. Discussion

Five non-*Homoneura* Homoneurinae genera were included in the present study intended to be an outgroup, but deeply nested into *Homoneura* based on four synapomorphies on the wing: spot on crossvein r-m absent (53:1), wing 2nd (between R_1_ and R_2+3_) section/3rd (between R_2+3_ and R_4+5_) section < 3X (60:1), area of transparent area or light-yellow area of wing/area of wing spot area ≤ 1 (49:1), and area of transparent or light-yellow area above wing R_4+5_/area of wing spot above R_4+5_ ≤ 1 (50:1). The monophyly of genus *Homoneura*, therefore, was not supported by our analysis. 

Sasakawa proposed the combination of pre-sutural dorsocentral seta absent and post-sutural dorsocentral seta three (dorsocentral setae 0 + 3) as the synapomorphy of subgenus *Homoneura* [21]. We found that pre-sutural dorsocentral seta present (32:1) was a synapomorphy of *Euhomoneura*, and pre-sutural dorsocentral seta absent was pleiomorphic and found in the remainder of the included subgenera. Multiple statuses were proposed in the present study regarding the number and location of post-sutural dorsocentral setae. Post-sutural dorsocentral setae three, firstst post-sutural dorsocentral seta behind the transversal suture (33:0) is plesiomorphic and found in non-*Euhomoneura Homoneura* subgenera, whereas post-sutural dorsocentral setae three, first post-sutural dorsocentral seta in the transversal suture (33:1) occurs in *Noonamyia umbrellata* and *Cestrotus liui*; post-sutural dorsocentral setae two (33:2) is a synapomorphy of *Euhomoneura*. Therefore, dorsocentral setae 0 + 3 cannot be an effective synapomorphy to support the monophyly of subgenus *Homoneura.*

The non-monophyly of the genus *Homoneura* and the subgenus *Homoneura* was also detected based on molecular data. Shi et al. obtained two mitochondrial genes (COI, 16S-rRNA) and two nuclear genes (Elongation factor 1-α, 28S-rRNA), and reconstructed gene trees using four different methods (NJ, ME, MP, and ML). Among 16 resulted gene trees, the monophyly of genus *Homoneura* and subgenus *Homoneura* was never recovered (they always had *Minettia*, or *Sapromyza*, or both nested) [14]. 

The sister relationship between the subgenera *Chaetohomoneura* and *Neohomoneura* was also suggested by Sasakawa [21]. Sasakawa proposed two potential ‘plesiomorphic’ characters to support *Chaetohomoneura* + *Neohomoneura*, among which mid tibia with posterior seta (44:1) was found to be the synapomorphy by our current analysis. Another character, acrostichal seta no less than seven rows (34:0) was found to be homoplastic in multiple species of the subgenera *Minettioides* and *Homoneura.* To alleviate the difficulty of distinguishing these two subgenera, Shi et al. summarized three characters: *Chaetohomoneura* with two supra-alar setae, four strong apical ventral setae on mid tibia, and posterior ventral setae often present in mid femur; *Neohomoneura* with one supra-alar seta, three strong apical ventral setae on mid tibia, and posterior ventral setae often absent from mid femur [22]. We found that additional characters could help distinguish these two subgenera: *Chaetohomoneura* with posterior ventral seta in mid femur present, and *Neohomoneura* with wing with four spots, separately on tip of R_2+3_, R_4+5_, M_1,_ and crossvein dm-cu.

Sasakawa suggested intra-alar seta absent as a synapomorphy to support the monophyly of subgenus *Euhomoneura* + subgenus *Homoneura* [21]. Based on the present analysis, we found that the character is plesiomorphic on our tree and present in all species of non-*Minettioides* Chinese subgenera of the genus *Homoneura*. The synapomorphies support the monophyly of subgenera *Euhomoneura,* which is consistent with Sasakawa (one pre-sutural dorsocentral setae (32:1) and two post-sutural dorsocentral setae (33:2)).

## 5. Conclusions

This study presents the first morphological phylogeny of *Homoneura*, based on 105 characters of adults and 24 species representing all five subgenera of *Homoneura* recorded from China, underpinning our understanding of the phylogenetic relationships in the group.

Our results show that the monophyly of the genus *Homoneura* and subgenus *Homoneura* is not supported. Additionally, the results show that the monophyly of the subgenera *Euhomoneura* and *Neohomoneura* is supported, as well as the sister relationship between the subgenera *Chaetohomoneura* and *Neohomoneura*. Due to our regional taxon sampling, our result are premature to propose a new classification for genus or subgenus *Homoneura*, but we discovered the urgent need to revise this diverse group. Future studies with global taxon sampling, morphological evidence from multiple life stages, and molecular data are needed to reconstruct the phylogeny of Homoneurinae and revise the classification. 

## Figures and Tables

**Figure 10 insects-13-00665-f010:**
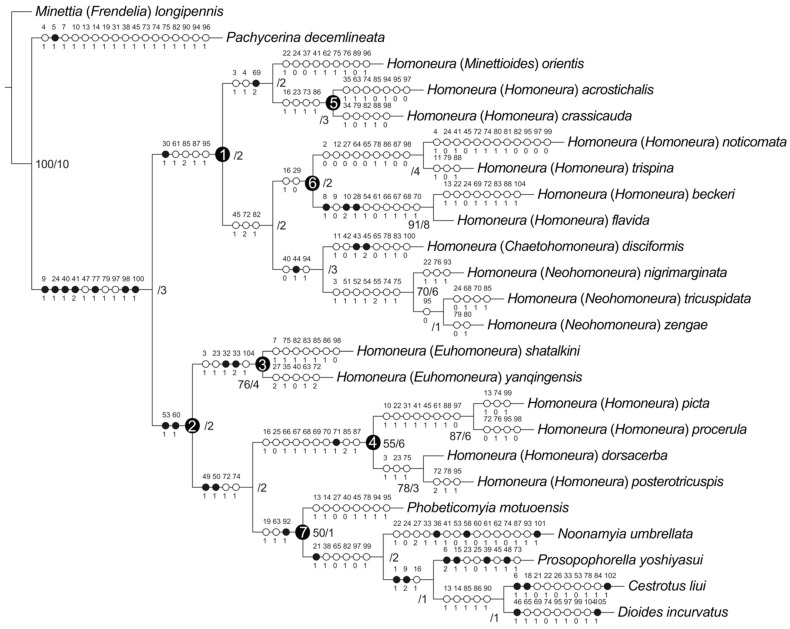
Phylogenetic relationships of *Homoneura.* Synapomorphies are marked by filled circles and homoplasious characters by open circles. The character numbers and states are placed above and below the circles, respectively. Bremer support: left, bootstrap values > 50%; right, Bremer support values.

## Data Availability

Data is contained within the article and Appendix A.

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
