# Peer review of "Phylogeny of the Chinese Subgenera of the Genus Homoneura (Diptera, Lauxaniidae, Homoneurinae) Based on Morphological Characters"

_insects, 2022, doi:10.3390/insects13080665_

Round 1

Reviewer 1 Report

The manuscript insects-1823677, entitled “Phylogeny of the Chinese subgenera of the genus Homoneura (Diptera, Lauxaniidae, Homoneurinae) based on morphological characters” by Kong et al., presents a phylogenetic study of the Chinese species of Homoneura. The text is well-written, and the figures are of high quality, thus contributing to resolve the phylogenetic relationships of Lauxaniidae. The manuscript can be accepted for publication after a minor revision.

Citation: The surnames and the initials of the first names are confused. For the Chinese authors such as the first author, Chaoyang Kong, “Kong” is the family name, and “Chaoyang” is the first name.

L44: “(Lauxaniidae, Homoneurinae)” seems redundant, since Lauxaniidae appears at the end of this sentence.

L46: “more than species are recorded”. Evidently, a numeral is missing before “species”.

L69: To macerate insect specimens, we never used saturated NaOH. Please explain why saturated?

L74: “drowned” is mis-spelling, please use the correct form of past participle for the word draw.

L83-86: This sentence is awkward, for it has two verbs (were obtained, and were numerically coded).

L88: Please insert a period at the end of the sentence.

L241: “spex”, this word looks like a misspelling. It could be “apex”?

L261-270, and others: For character encoding, the authors had better to use different names for different characters.

L326: “including” should be “includes”.

L381-387: seta and setae are used chaotically.

L387: Please modify “subgenera” to “subgenus”.

L454: ““Physoclypeus farinosus” is a scientific name, should be in italics.

L480: “china” should be “China”.

Reviewer 2 Report

This study present by the authors is a well-treated and illustrated morphological phylogeny of the genus Homoneura. It is a great contribution to the systematics of the family Lauxaniidae. 

It was a bit confusing the figure arrangements at first. Perhaps this might be explained in the methodology and or as a sentence previously to the figures.  I would have like the authors to have included few adults photographs (habitus) to show the diversity of the specimens included in this study.

This study presents a well-treated and illustrated morphological phylogeny of the genus Homoneura. It is a great contribution to the systematics of the family Lauxaniidae. The discussion of the SubfamilyHomoneurinae is well delivered. 

In line 428 www.mdpi.com/xxx/s1, Table S1: The species studied, Table S2: Morphological dataset used for the analysis of the phylogeny. 

This site presented with an error 404 File not found, and I could not see it. 

I found minor errors in the following lines:

131  the “length” of the antennomere?

280 instead “Globular”
